# Exploring Molecular Signatures Associated with Inflammation and Angiogenesis in the Aqueous Humor of Patients with Non-Proliferative Diabetic Retinopathy

**DOI:** 10.3390/ijms26136461

**Published:** 2025-07-04

**Authors:** Víctor Alegre-Ituarte, Irene Andrés-Blasco, David Peña-Ruiz, Salvatore Di Lauro, Sara Crespo-Millas, Alessio Martucci, Jorge Vila-Arteaga, María Dolores Pinazo-Durán, David Galarreta, Julián García-Feijoo

**Affiliations:** 1Cellular and Molecular Ophthalmo-Biology Group (GIUV 13/152), Department of Surgery, Faculty of Medicine and Odontology, University of Valencia, 46010 Valencia, Spain; valegrei@csi.cat (V.A.-I.); irene.andres@fisabio.es (I.A.-B.); 2Ophthalmic Research Unit “Santiago Grisolía”/FISABIO, 46017 Valencia, Spain; 3Ophthalmology Service, Complex Hospitalari Universitari Moisès Broggi, 08970 Sant Joan Despí, Barcelona, Spain; 4Spanish Network of Inflammatory Diseases and Immunopathology of Organs and Systems (REI-RICORS; RD24/0007/0004), Institute of Health Carlos III, Ministry of Science and Innovation, 28029 Madrid, Spain; 5Ophthalmology Service, University Clinic Hospital, 47003 Valladolid, Spain; d.pena-ruiz@ioba.es (D.P.-R.); dilauro@gmail.com (S.D.L.); sara.crespo@gmail.com (S.C.-M.); galarreta@hotmail.com (D.G.); 6Ophthalmology Service, University of Roma Tor Vergata, 00133 Roma, Italy; martucci@med.uniroma2.it; 7Ophthalmology Service, University and Polytechnic Hospital La Fe, 46026 Valencia, Spain; vila.arteaga@innova.es; 8Ophthalmology Service, University Clinic Hospital San Carlos, 28040 Madrid, Spain; jgarciafeijoo@gmail.com; 9Institute of Health Research, San Carlos Clinic Hospital (IdISSC), 28040 Madrid, Spain

**Keywords:** type 2 diabetes mellitus, non-proliferative diabetic retinopathy, aqueous humor, inflammation, angiogenesis, multiplex bead-based immunoassay, biomarkers, retina

## Abstract

Type 2 diabetes mellitus (T2DM) is a major public health concern that significantly increases the risk of diabetic retinopathy (DR), a leading cause of visual impairment worldwide. This study aimed to identify molecular markers of inflammation (INF) and angiogenesis (ANG) in the aqueous humor (AH) of patients with non-proliferative diabetic retinopathy (NPDR). We conducted an observational, multicenter, case–control study including 116 participants classified into T2DM with NPDR, T2DM without DR, and non-diabetic controls (SCG) undergoing cataract surgery. AH samples were collected intraoperatively and analyzed for 27 cytokines using multiplex immunoassay. Eighteen immune mediators were detected in AH samples, and several were significantly elevated in the NPDR group, including the interleukins (IL) -1β, -6, -8, -15, -17, as well as the granulocyte–macrophage colony stimulating factor (GM-CSF), basic fibroblast growth factor (bFGF), interferon gamma-induced protein (IP-10), macrophage inflammatory protein 1 beta (MIP-1b), monocyte chemoattractant protein-1 (MCP-1), regulated on activation, normal T cell-expressed and -secreted protein (RANTES), and the vascular endothelial growth factor (VEGF). These molecules are involved in retinal INF, blood–retinal barrier breakdown, and pathological neovascularization. Our findings reveal a distinct pro-INF and pro-ANG profile in the AH of NPDR patients, suggesting that these cytokines may serve as early diagnostic/prognostic biomarkers for DR. Targeting these molecules could provide novel therapeutic strategies to mitigate retinal damage and vision loss in diabetic patients.

## 1. Introduction

Diabetic retinopathy (DR) is the leading global cause of visual impairment among working-age adults and one of the most common microvascular complications of type 2 diabetes mellitus (T2DM) [1,2]. As the global prevalence of diabetes rises, the burden of DR increases proportionately, posing a significant challenge for healthcare systems and highlighting the urgent need for improved diagnostic, preventive, and therapeutic strategies [3]. Epidemiological studies estimate that approximately 27% of diabetic patients develop DR, with 4.2% progressing to vision-threatening stages [3,4]. Risk factors for DR include poor glycemic control, disease duration, hypertension, dyslipidemia, and genetic predisposition [5]. The Early Treatment Diabetic Retinopathy Study (ETDRS) classification divides DR into non-proliferative (NPDR) and proliferative (PDR) stages, with diabetic macular edema (DME) potentially arising at any stage [6].

From a pathophysiological standpoint, chronic hyperglycemia promotes oxidative stress (OXS), leading to the overproduction of reactive oxygen species (ROS) and formation of advanced glycation end-products (AGEs) [7]. These, in turn, activate multiple signaling cascades, including the protein kinase C (PKC), polyol, hexosamine, and mitogen-activated protein kinase (MAPK) pathways [8,9,10,11], all contributing to endothelial dysfunction and blood–retinal barrier (BRB) breakdown. Inflammatory and angiogenic mediators such as vascular endothelial growth factor (VEGF), basic fibroblast growth factor (bFGF), interleukin (IL)-6, tumor necrosis factor alpha (TNF-α), intercellular adhesion molecule 1 (ICAM-1), vascular adhesion molecule 1 (VCAM-1), monocyte chemoattract protein 1 (MCP-1), and others are upregulated [7,9,10,12,13], triggering microaneurysm formation, ischemia, neovascularization, and neuronal apoptosis. Apoptotic pathways involving caspase-3 (CAS3) and poly adenyl ribose polymerase 1 (PARP-1) are also activated [7,14,15,16]. Prior studies by our group demonstrated elevated levels of IL-1β, IL-6, TNF-α, VEGF, CAS3, and PARP1 in the plasma, tears, and vitreous of patients with NPDR, PDR, and/or diabetic macular edema (DME) [7,9,17].

Recent research underscores the importance of epigenetic regulation, particularly microRNAs (e.g., miR-10a-5p, miR-15b-5p), which modulate OXS, apoptosis, and ANG by targeting the B-cell lymphoma 2 like 2 (*BCL2L2*) gene, and the gene encoding the nuclear phosphoprotein of 53 kilo Dalton (kD) (*TP53*) [18,19]. In parallel, cytokines such as IL-15, IL-17, IP-10, MCP-1, macrophage inflammatory protein-1-beta (MIP-1β), and granulocyte macrophage–colony stimulating factor (GM-CSF) have been implicated in DR pathogenesis [20,21,22,23]. Furthermore, IL-15 and IL-17 promote chronic INF and BRB disruption [24,25], while interferon γ-induced protein 10 kDa (IP-10), also known as C-X-C motif chemokine 10 (CXCL10), and chemokine receptor 3 (CXCR3) facilitate immune cell recruitment and microglial activation [26,27,28]. MCP-1 and MIP-1β further drive monocyte infiltration and retinal INF, correlating with DR severity [29,30]. In addition, bFGF cooperates with VEGF in driving retinal pathologic ANG and DME development [31,32,33], and GM-CSF enhances microglial activation, contributing to neurovascular damage [22,23]. 

Despite current interventions—including anti-VEGF therapies, laser photocoagulation, and vitreoretinal surgery—no definitive cure exists for DR, and vision loss is too often irreversible [7]. A deeper understanding of intraocular molecular mechanisms is essential in developing earlier diagnostic tools and targeted therapies. In this context, this study aimed to characterize the INF and ANG molecular profiles in the aqueous humor (AH) of T2DM patients with NPDR. By identifying specific cytokines and growth factors involved in early retinal alterations, we seek to define potential diagnostic–prognostic biomarkers, as well as to explore therapeutic targets to prevent DR progression and associated visual loss.

Previous studies have investigated proinflammatory (pro-INF) and proangiogenic (pro-ANG) mediators in plasma or vitreous body (VB) samples from diabetic patients, often in advanced stages of retinopathy, or under treatment. Vitreous studies, while informative, are invasive and typically limited to patients undergoing vitrectomy. Plasma cytokine levels, on the other hand, may not accurately reflect intraocular conditions. Few studies have comprehensively profiled cytokines in the aqueous humor (AH), particularly in the early stages of DR. Our study addresses this gap by analyzing 27 immune-related molecules in AH from patients with NPDR, that were treatment-naïve and without systemic comorbidities, using a sensitive multiplex platform. This allows for the identification of local immune signatures that may contribute to early disease mechanisms and serve as potential biomarkers or therapeutic targets.

## 2. Results

### 2.1. Study Cohort and Group Assignment

A total of 166 volunteers were initially recruited. After an initial screening and double evaluation, 138 participants met the inclusion criteria. Following personal interviews, full ophthalmological evaluation, cataract surgery, and successful sampling procedures, 116 individuals were included for final statistical analysis. The study population was divided as follows: 80 individuals with T2DM, of whom 38 had non-proliferative diabetic retinopathy (T2DM+DR) and 42 had no DR (T2DM–DR), and 36 non-diabetic participants who formed the surrogate control group (SCG). Participant attrition was attributed to ineligibility, incomplete data, and sample quality issues.

### 2.2. Sociodemographic and Clinical Characteristics

The mean age of participants was 59 ± 5 years in the SCG group, 62 ± 5 years in the T2DM–DR group, and 65 ± 7 years in the T2DM+DR group. No significant differences were observed in age or sex distribution across the study groups. Duration of diabetes and family history were only assessed among diabetic participants. A summary of characteristics is presented in Table 1.

### 2.3. Ophthalmological Evaluation

All participants underwent a comprehensive ophthalmic assessment including BCVA, IOP, and SD-OCT scanning. The results are summarized in Table 2. BCVA was significantly reduced in the T2DM + DR group compared to the SCG (*p* = 0.005), consistent with retinopathy-associated visual deterioration. IOP values showed a trend toward elevation in diabetic patients, but differences were not statistically significant. Central subfield foveal thickness (CSFT), as measured by SD-OCT, was notably higher in patients with DR, consistent with early retinal thickening. SD-OCT scans were exported as 50 × 50 superpixel arrays aligned to the central 20° field and evaluated by two masked ophthalmologists.

### 2.4. Biochemical Profile

Fasting glycemia, glycated hemoglobin (HbA1c), total cholesterol (T Chol), and triglycerides (TRGs) were assessed in all participants. Diabetic groups showed significantly elevated levels in all four parameters compared to SCG, confirming systemic metabolic dysregulation. Complete values are presented in Table 3. 

### 2.5. Molecular Profiling of Aqueous Humor

The AH samples from all study participants were analyzed using a bead-based multiplex immunoassay platform (Luminex®) to quantify a panel of 27 immune mediators involved in inflammation and angiogenesis. This panel included five anti-inflammatory cytokines, twelve pro-inflammatory mediators, six additional modulators, and four vascular-related growth factors (full list in Methods).

Each analyte’s concentration was compared against its minimal detectable concentration (MDC), and only molecules that were reliably quantified (i.e., measurable in a majority of samples) were included in subsequent statistical analysis. Based on this criterion, 18 of the 27 mediators (66.6%) were detectable in the aqueous humor of the study participants, while 9 molecules (IL-4, IL-5, IL-9, Eotaxin, G-CSF, IFN-γ, MIP-1α, TNF-α, and PDGF) were excluded due to having levels below the MDC in most or all samples.

Table 4 summarizes the MDC values and the percentage of samples in which each cytokine was detectable. Molecules undetectable in 100% of samples were shaded in gray (in the original manuscript format) and excluded from statistical comparison.

This quality control step ensured that downstream group comparisons were based on biologically relevant and technically reliable measurements.

### 2.6. Expression Patterns and Group Comparisons

A comparative analysis of the 18 detectable cytokines, chemokines, and growth factors revealed significant differences in aqueous humor (AH) profiles between T2DM vs non-diabetic participants, as well as between T2DM individuals +NPDR vs -NPDR. Full quantitative results and statistical comparisons are presented in Table 5.

Several proINF mediators, including the ILs -1β, -6, and -8, showed significantly elevated levels in the AH of T2DM participants compared to controls, with further increases observed in the presence of NPDR. For example, IL-1β increased from 1.64 ± 0.23 pg/mL in SCG to 7.0 ± 2.09 pg/mL in the T2DM + DR group (*p* = 0.0014).

Among the angiogenesis-related molecules, VEGF, bFGF, and GM-CSF were markedly elevated in T2DM + DR patients. VEGF rose from 65.26 ± 20.8 pg/mL in SCG to 118.4 ± 30.16 pg/mL in the T2DM + DR group (*p* = 2.71 × 10^−8^), while GM-CSF increased from 50.58 ± 9.5 pg/mL in controls to 91.32 ± 11.16 pg/mL (*p* = 2.27 × 10^−7^).

Strong differences were also observed for chemokines involved in leukocyte recruitment and microglial activation. IP-10, MCP-1, and MIP-1β all exhibited significantly higher concentrations in the T2DM + DR group compared to both T2DM–DR and SCG. For instance, IP-10 levels were 51.68 ± 28.6 pg/mL in SCG and 205.31 ± 92.18 pg/mL in T2DM + DR (*p* = 1.60 × 10^−13^).

Some immune mediators, including IL-1Ra, IL-15, IL-17, and RANTES, also showed increased concentrations in diabetic groups, though differences between T2DM–DR and T2DM + DR did not always reach statistical significance.

Conversely, cytokines such as the ILs -2, -7, -10, -12, and -13, did not show significant differences across the study groups, suggesting the selective activation of immune-inflammatory pathways in NPDR. 

The eleven molecules with the highest statistical discrimination between groups—ILs -1Ra, -6, -15, -17, bFGF, GM-CSF, IP-10, MIP-1β, MCP-1, RANTES, and VEGF—appear to form a distinct intraocular proINF and proANG profile characteristic of early DR.

The most relevant findings are visualized in the boxplots of the 18th most significantly elevated molecules in the AH of the study participants, with respect to the comparisons between groups and subgroups (the latter regarding the presence or not of NPDR), as shown in Figure 1. 

## 3. Discussion

The DR remains one of the most frequent microvascular complications of diabetes, and a leading cause of visual impairment worldwide. Despite considerable advances in screening and treatment, there is still a critical need for earlier detection and for the identification of intraocular biomarkers capable of reflecting subclinical retinal inflammation and vascular dysfunction. In this study, we investigated the molecular profile of the aqueous humor (AH) in T2DM patients with and without non-proliferative diabetic retinopathy, comparing them to a non-diabetic control group undergoing cataract surgery. Our aim was to detect inflammatory and angiogenic signals in the anterior segment of the eye that may help characterize early stages of DR and contribute to future biomarker development.

The final analysis included 116 participants, distributed into three well-defined groups: T2DM + DR, T2DM–DR, and non-diabetic controls (SCG). Sociodemographic characteristics such as age and sex were balanced across groups, and systemic metabolic profiles (glycemia, HbA1c, total cholesterol, triglycerides) were, as expected, significantly altered in diabetic patients. This establishes a clinical context consistent with previous epidemiological studies showing that poor metabolic control correlates with retinal microvascular disease progression [34].

Our ophthalmologic evaluation revealed functional and anatomical differences between groups. BCVA was significantly reduced in the T2DM + DR group, consistent with early visual dysfunction. CSFT, measured by SD-OCT, was elevated in diabetic individuals and highest among those with DR, supporting the presence of early retinal thickening and possible subclinical edema. These findings align with prior observations of increased foveal thickness and macular volume in NPDR [35,36,37]. Although intraocular pressure (IOP) did not differ significantly across groups, a slight upward trend in diabetics may reflect subtle changes in aqueous humor dynamics associated with vascular compromise or early ciliary body dysfunction.

The multiplex immunoassay allowed us to detect 18 of 27 analyzed cytokines and growth factors in AH samples. This detection rate (66.6%) validates the sensitivity of the Luminex® platform in aqueous fluid and emphasizes the biological relevance of the selected mediators. Of the 27 molecules analyzed in the human cytokine panel by the bead-based multiplex immunoassay used herein, only 9 were excluded from the statistical processing, specifically those that fell below the MDC in at least 50% of the AH samples: IL-4, IL-5, IL-9, Eotaxin, G-CSF, IN-γ, MIP-1α, TNF-α, and PDGF. Their absence or undetectable levels may be attributed to a combination of technical and biological factors. Also, unknown factors were taken into consideration. Technically, certain analytes have relatively high MDC within the bead-based multiplex immunoassay, limiting sensitivity for low-abundance targets. Biologically, these cytokines may have limited intraocular expression in non-inflamed eyes, as well as during early DR stages. Prior aqueous humor studies have reported similar findings, suggesting that these molecules may play a minor or localized role in anterior segment immune activity in NPDR [38,39]. It has also been reported that, specifically, the IL-1β, TNF-α, and IF-γ expression levels highly varied across assays [40].

In this regard, it is critical to carefully consider the high intra-individual variability of the measurements, as well as the elevated discrepancy between these types of analytical platform, drawing from the literature, which may lead to poor agreement regarding the identification of reliable biomarkers for ocular diseases, including DR. Therefore, there is an urgent need for high-performance, sensitive, precise and validated platforms for the proper discovery of biomarkers of INF and ANG for translation into clinical diagnostics, as well as for designing new therapies [40,41].

Finally, many researchers have compared blood cytokine levels in a wide variety of diseases with those of healthy controls, including our group [42,43]; however, very few publications have demonstrated that these cytokines remain stable in healthy individuals through time periods of days, weeks, months and years, which is also an important point to consider in these studies [44,45].

Among the molecules detected, a clear upregulation of pro-inflammatory and pro-angiogenic factors was observed in diabetic patients, particularly those with DR. IL-1β and IL-6 were markedly elevated in the T2DM + DR group, reinforcing their central role in initiating and perpetuating retinal inflammation. IL-1β has been implicated in leukostasis and endothelial barrier breakdown, while IL-6 contributes to increased vascular permeability and Müller glia activation [46,47]. IL-8, a chemokine with dual roles in inflammation and angiogenesis, also showed a progressive increase from SCG to T2DM–DR and T2DM + DR, suggesting that its concentration may reflect retinopathy severity. Similarly, MCP-1 and MIP-1β—chemokines known to attract monocytes and promote leukocyte adhesion—were significantly upregulated in the diabetic groups, consistent with previous reports linking them to leukostasis and capillary occlusion in diabetic retinas [48,49].

VEGF, a well-established driver of pathological neovascularization in DR, was significantly increased in T2DM + DR patients, confirming its key role even during non-proliferative stages. The increase in VEGF, accompanied by elevated levels of bFGF and GM-CSF, suggests the existence of a pro-angiogenic milieu in the anterior chamber of diabetic eyes [48,49]. 

The bFGF not only promotes endothelial cell proliferation but also supports extracellular matrix remodeling and pericyte migration, all of which are relevant to diabetic microangiopathy [50,51].

Notably, GM-CSF and IP-10, which are less frequently reported in aqueous humor studies, showed marked elevations in T2DM + DR. These mediators are involved in microglial activation and the recruitment of Th1-type lymphocytes, indicating contributions from both innate and adaptive immune responses in the early phases of DR [52,53]. IP-10 (CXCL10), in particular, showed one of the largest fold changes between groups, reinforcing its emerging role as a biomarker of retinal ischemia and inflammation [54,55].

Interestingly, not all cytokines traditionally associated with inflammation were elevated. IL-2, IL-12, IL-13, and IL-10 did not differ significantly across the three groups. This selective activation pattern suggests that DR is associated with specific inflammatory axes, rather than a global cytokine storm. Moreover, the presence of elevated IL-1Ra in T2DM + DR, an anti-inflammatory receptor antagonist, may reflect a compensatory response to chronic IL-1β activity, as proposed in other models of neuroinflammation [56,57]. Elevated IL-15 and IL-17 were also detected in diabetic groups, although the differences between T2DM + DR and T2DM–DR were less pronounced. These findings align with recent reports identifying these molecules as contributors to BRB disruption and T cell-mediated damage in the retina [58,59].

Our data support the concept that the aqueous humor, although anatomically anterior, reflects posterior segment inflammation and vascular stress. The presence of upregulated mediators in the AH of diabetic eyes suggests the active diffusion or production of these molecules from ciliary epithelium, retinal circulation, or infiltrating leukocytes. This observation strengthens the rationale for using AH as a minimally invasive biomarker source for early retinal disease, complementing OCT-based imaging and systemic biomarkers [60,61,62,63].

Furthermore, the ability to differentiate the T2DM–DR from the T2DM + DR based on AH cytokine profiles (e.g., IL-1β, bFGF, GM-CSF, VEGF) suggests potential utility for early risk stratification and monitoring treatment responses in clinical practice [60,61,62,63,64].

Several limitations should be acknowledged. First, the study’s cross-sectional design limits interpretations of causal relationships or longitudinal dynamics. Second, the group of non-diabetic patients undergoing cataract surgery was named the surrogate comparative-control group (SCG). While cataracts are not considered pathological processes per se, and complicated cataracts have been excluded from this work, it has been contemplated that the surgical maneuvers, and the intense light beams of the microscope, may theoretically influence the integrity and composition of anterior segment tissues and the AH. Previous studies suggest that cytokine levels in the AH from cataract patients without systemic or ocular comorbidities are comparable to those of healthy eyes [22,23,60,61,62,63,64]. Nonetheless, this remains a potential source of bias that warrants caution in interpretation. Third, although the Luminex platform offers multiplex capability, its sensitivity may have excluded the detection of certain low-abundance cytokines. Moreover, the limited volume of AH samples precluded complementary analyses such as proteomics or cell population studies. Finally, variability in OCT machines and measurement protocols, although minimized through standardization, may still influence thickness metrics [40,41]. The most relevant noninvasive ancillary probing for the diagnosis of sight-threatening eye diseases, such as DR, is the SD-OCT. However, it has been reported that measurements from multi-vendor OCT devices are not fully interchangeable [65]. Macular thickness measurements can be affected by the OCT platform used, particularly due to differences in segmentation algorithms and retinal boundary definitions between Heidelberg Spectralis and Zeiss Cirrus systems. To mitigate this, acquisition protocols were standardized, and all scans were reviewed by the same retina specialist to ensure consistency and exclude artifacts. Nevertheless, residual inter-platform variability cannot be fully excluded and represents a potential source of measurement bias. Novel segmentation algorithms are urgently needed to allow ophthalmologists to test different systems with a general standardized reference.

Further research is needed to evaluate whether the identified molecular signatures persist in the vitreous and correlate with retinal imaging biomarkers such as OCT angiography or microperimetry. In addition, prospective studies could explore whether the targeted inhibition of mediators like IL-6, MCP-1, or GM-CSF might slow DR progression. Combining immunological profiles with genetic and epigenetic data could also help identify susceptible phenotypes and inform personalized treatment strategies. The development of predictive models integrating these data layers may ultimately guide risk stratification and early intervention in diabetic eye disease.

In conclusion, this study identifies a distinct profile of intraocular inflammation and angiogenesis in the aqueous humor of patients with early diabetic retinopathy. Elevated levels of IL-1β, IL-6, IL-8, VEGF, bFGF, GM-CSF, IP-10, MCP-1, and MIP-1β highlight a multifactorial process involving immune activation, vascular remodeling, and neuroretinal stress. These findings provide a foundation for developing biomarker-guided strategies to detect and manage DR before irreversible vision loss occurs.

## 4. Materials and Methods

### 4.1. Study Design

This multicenter, observational, case–control study was conducted between 2023 and 2025 at the University Clinical Hospital of Valladolid, the University and Polytechnic Hospital La Fe in Valencia, and the University Clinical Hospital San Carlos in Madrid. The study aimed to evaluate inflammatory and angiogenic biomarkers in the AH of patients with T2DM, with or without NPDR, compared to non-diabetic SCG participants undergoing cataract surgery. The research protocol was approved by the Ethics Committee of University Hospital Dr. Peset, Valencia, Spain (protocol code: 42.22/6 May 2022), and all participants provided written informed consent.

Sample size estimation was conducted using the epicalc package in R software (version 4.2.2) [66]. The calculation was based on an expected medium effect size (Cohen’s d = 0.65) for intergroup differences in cytokine levels, with a two-sided significance level (α) of 0.05 and a statistical power (1–β) of 0.80. Under these assumptions, a minimum of 32 participants per group were required to detect statistically significant differences.

From an initial pool of 166 volunteers, 138 were selected for screening. After a complete clinical interview, ocular examination, cataract surgery, and biological sampling, a total of 116 individuals were included for final analysis: T2DM + NPDR (*n* = 38), T2DM–DR (*n* = 42), and SCG (*n* = 36). A schematic of the recruitment and selection process is provided in Figure 2.

### 4.2. Inclusion and Exclusion Criteria

Eligible participants were adults aged 40 to 80 years with complete clinical records. Diabetic patients were required to have a confirmed diagnosis and accurate classification of NPDR based on the ICO severity scale [67]. Exclusion criteria included other types of diabetes or DR, ocular or systemic comorbidities, eye or laser surgery in the previous 12 months, or inability to provide informed consent. The SCG consisted of healthy individuals attending ophthalmology clinics for cataract diagnosis and phacoemulsification surgery. Surgery was indicated due to advanced lens opacification, decreased visual acuity, or glare sensitivity affecting quality of life. The diagnosis of NPDR was based on the International Council of Ophthalmology (ICO) severity scale [67]. Only patients without macular involvement were included (Table 6).

### 4.3. Ophthalmological Examination

All participants underwent a standardized bilateral ophthalmic examination. Visual acuity was evaluated using the logarithm of the minimum angle of resolution (LogMAR) scale for best corrected visual acuity (BCVA). The anterior segment was examined by slit-lamp biomicroscopy using either the SL 900 ImageNet (Topcon, Barcelona, Spain) or the B900 slit lamp (Haag-Streit, Switzerland). Intraocular pressure (IOP) was measured in each eye using Goldmann applanation tonometry (AT 900, Haag-Streit, Switzerland).

Posterior segment evaluation included color fundus photography (CFP; ImageNet, Topcon, Barcelona, Spain) and macular analysis using spectral domain optical coherence tomography (SD-OCT). Two SD-OCT platforms were used: Spectralis (Heidelberg Engineering GmbH, Heidelberg, Germany) and Cirrus HD-OCT (Carl Zeiss Meditec, Madrid, Spain). The OCT parameters evaluated included central subfield foveal thickness (CSFT) and cube average thickness (CAT). SD-OCT scans were aligned and exported as 50 × 50 superpixel arrays centered on the central 20° of the macula [40,41,65]. Images were independently reviewed by three retina specialists (V.A.-I., S.D.L., S.C.-M., M.D.P.-D.).

NPDR grading was performed according to the International Council of Ophthalmology (ICO) severity scale [67], based on the presence and extent of microaneurysms, retinal hemorrhages, venous beading, and intraretinal microvascular abnormalities (IRMAs). Participants with macular involvement were excluded. Worsening was defined as the progression of any of the aforementioned funduscopic signs during the study course.

### 4.4. Sampling Procedures 

#### 4.4.1. Blood Samples and Biochemical Testing

Blood samples were obtained from the antecubital vein of all participants in fasting conditions, at 08:00 a.m. on the day scheduled for cataract surgery. Blood was collected into 4.5 mL tubes containing ethylenediaminetetraacetic acid (EDTA) or sodium citrate as anticoagulants (Becton Dickinson, Auckland, New Zealand). One EDTA tube (purple cap) per participant was used for biochemical analysis at the Clinical Analysis Departments of the participating centers.

Automated chemistry analyzers were employed for glycemic and lipid profiling: (1) the Architect c8000 analyzer (Abbott Laboratories, Abbott Park, IL, USA) and (2) the Arkray AU 4050 analyzer (Arkray Global Business Inc., Kyoto, Japan). Measurements included fasting plasma glucose, glycated hemoglobin (HbA1c), total cholesterol (T Chol), and triglycerides (TRGs). All procedures were carried out under the supervision of a clinical biochemistry specialist.

#### 4.4.2. Aqueous Humor Sampling

AH samples were obtained at the onset of cataract surgery through corneal paracentesis. The surgeon gently aspirated the AH using a Rycroft cannula attached to a 2 mL syringe, collecting 100–120 μL of fluid from the anterior chamber. Each sample was immediately transferred to labeled cryotubes and stored in insulated freezing boxes at −80 °C. Transportation to the central research laboratory was performed under strict cold chain conditions.

Upon arrival, samples were inspected, registered, and stored again at −80 °C until molecular analysis. No dilution was applied to AH samples at any step of the process.

Molecular profiling of the AH was performed using a Luminex®-based bead multiplex immunoassay platform (Bio-Plex Pro™ Human Cytokine 27-plex Assay; Bio-Rad Laboratories, Hercules, CA, USA). The following 27 analytes were simultaneously measured:Anti-inflammatory ILs: IL-1Ra, IL-4, IL-5, IL-10, IL-13Pro-inflammatory cytokines: IL-1β, IL-2, IL-6, IL-12, IL-17, IL-18, IFN-γ, TNF-αChemokines: IP-10, MCP-1, MIP-1α, MIP-1β, RANTES, EotaxinOthers: IL-15, MIF, G-CSF, GM-CSFAngiogenic and hematopoietic factors: IL-7, bFGF, PDGF, VEGF

All assays were performed in duplicate and according to the manufacturer’s instructions. Briefly, samples were incubated with magnetic beads coated with analyte-specific capture antibodies. After several washing steps to remove unbound material, biotinylated detection antibodies were added and incubated for 30 minutes. Streptavidin–phycoerythrin was used as a fluorescent reporter, binding to the detection antibodies. Fluorescence intensity was acquired using the Bio-Plex® suspension array system. Each cytokine/chemokine was identified by bead color and quantified via reporter signal intensity. Concentrations were automatically calculated using Bio-Plex Manager software based on a standard curve generated from recombinant protein standards. Results were normalized to the total protein content of each sample and expressed in pg/mL. All samples (blood and AH) were analyzed in duplicate to ensure reproducibility. Final cytokine concentrations were reported as means ± standard deviation. Assay sensitivity was verified by determining the minimal detectable concentration (MDC) for each molecule, as per assay specifications. Values below MDC were excluded from comparative analysis, as reported elsewhere [9,17,19,36,38,48,49,68,69].

### 4.5. Statistical Analysis

All data were entered in Microsoft Excel and analyzed using R software (version 4.2.2; R Foundation for Statistical Computing, Vienna, Austria; https://www.R-project.org/) [70]. Normality was assessed using the Shapiro–Wilk test. Group comparisons were performed using Student’s t-test or Mann–Whitney U test, as appropriate. Correlations were analyzed using Pearson or Spearman coefficients. A *p*-value < 0.05 was considered statistically significant.

## 5. Conclusions

This study identifies a distinct inflammatory and angiogenic signature in the AH of patients with NPDR. Elevated levels of IL-1β, IL-6, VEGF, GM-CSF, and chemokines such as MCP-1 and IP-10 suggest a localized intraocular response in early diabetic retinal disease. These findings support the potential utility of aqueous biomarkers for early diagnosis, risk stratification, and therapeutic targeting in DR.

## Figures and Tables

**Figure 1 ijms-26-06461-f001:**
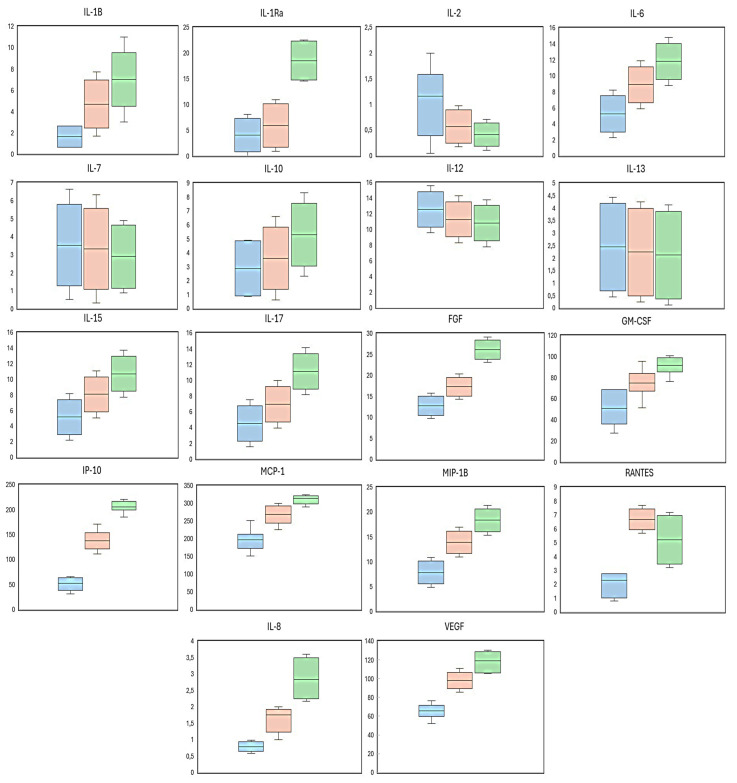
Levels of the pro-INF and pro-ANG mediators in the AH of the study participants displaying higher statistical significance among all study determinations. Boxplots color codes: blue corresponds to the SCG group; pink corresponds to the T2DM + NPDR group; and finally green corresponds to the T2DM-DR group. IL: interleukin; IL-1RA: the interleukin-1 receptor antagonist; bFGF: basic fibroblast growth factor; GM-CSF: granulocyte–macrophage stimulating factor; IP10: interferon gamma-induced protein 10 kDa; MCP-1: monocyte chemoattractant protein-1; MIP-1β: macrophage inflammatory protein-1 beta; RANTES: regulated on activation, normal T cell-expressed and -secreted; VEGF: vascular endothelial growth factor.

**Figure 2 ijms-26-06461-f002:**
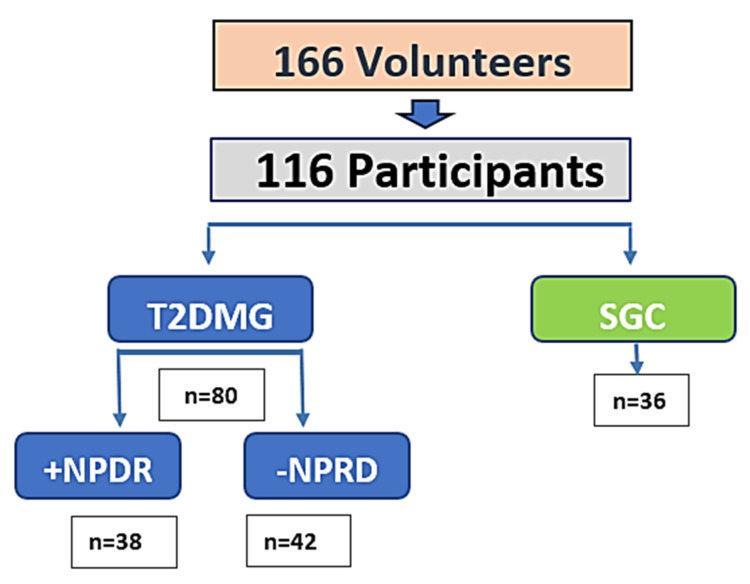
Flowchart of the recruitment characteristics and distribution of the study participants.

**Table 1 ijms-26-06461-t001:** Sociodemographic and clinical characteristics of the study participants.

Variables	SCG (*n* = 36)	T2DM–DR (*n* = 42)	T2DM + DR (*n* = 38)
Age (years)	59 ± 5	62 ± 5	65 ± 7
Sex (male/female)	16/20	22/20	26/12
T2DM duration (years)	–	15 ± 6	17 ± 8
T2DM familial History (%)	23%	58%	72%

SCG: surrogate comparative–control group; T2DM-DR: type 2 diabetics without retinopathy; T2DM + DR: type 2 diabetics with retinopathy.

**Table 2 ijms-26-06461-t002:** Ophthalmological parameters of the study participants.

Eye Parameters	SCG (*n* = 36)	T2DM–DR (*n* = 42)	T2DM + DR (*n* = 38)
BCVA LogMAR RE	0.09	0.30	0.44
BCVA LogMAR LE	0.08	0.35	0.52
IOP RE (mm Hg)	14 ± 3	16 ± 3	17 ± 2
IOP LE (mm Hg)	15 ± 2	16 ± 2	16 ± 2
CSFT RE (µm)	248 ± 20	295 ± 26	335 ± 34
CSFT LE (µm)	257 ± 21	275 ± 23	282 ± 25

BCVA: best corrected visual acuity, expressed in LogMAR units for each eye separately; RE: right eye; LE: left eye; IOP: intraocular pressure; CSFT: central subfield foveal thickness; SCG: surrogate comparative–control group; T2DM-DR: type 2 diabetics without retinopathy; T2DM + DR: type 2 diabetics with retinopathy.

**Table 3 ijms-26-06461-t003:** Biochemical profile of the study participants across the three study groups.

Variables	SCG (*n* = 36)	T2DM–DR (*n* = 42)	T2DM + DR (*n* = 38)
Basal Glycemia (mg/dL)	87 ± 8	124 ± 25	134 ± 32
HbA1c (%)	5.5 ± 0.3	6.8 ± 0.7	7.2 ± 1.1
Total Chol (mg/dL)	136 ± 34	186 ± 39	198 ± 42
Triglycerides (mg/dL)	76 ± 12	122 ± 15	130 ± 18

HbA1c: glycated hemoglobin; T Chol: total cholesterol; TRG: triglycerides; SCG: surrogate comparative–control group; T2DM–DR: type 2 diabetics without retinopathy; T2DM + DR: type 2 diabetics with retinopathy.

**Table 4 ijms-26-06461-t004:** Minimal detectable concentration (MDC; pg/mL) in the aqueous humor (AH) for each of the 27 assayed molecules by the bead-based immunoassay technique. Marked in gray are the 9 molecules that fell below the MDC in at least 50% of the AH samples.

Cytokine	MDC (pg/mL)	AH Sampling According to MDC (%)
**IL-1ra**	2.13	80
**IL-1β**	1.41	90
**IL-2**	0.56	78
**IL-4**	0.21	0
**IL-5**	2.50	0
**IL-6**	4.04	100
**IL-7**	1.71	76
**IL-8**	0.71	55
**IL-9**	0.50	0
**IL-10**	1.29	92
**IL-12**	7.76	100
**IL-13**	1.36	58
**IL-15**	3.42	95
**IL-17**	2.42	95
**bFGF**	7.28	85
**EOTAXIN**	2.10	0
**G-CSF**	2.05	0
**GM-CSF**	41.08	100
**IP10**	23.08	100
**IFγ**	1.70	0
**MIP1α**	1.10	0
**MIP1β**	5.76	100
**MCP1**	139.12	100
**RANTES**	0.87	92
**TNFα**	5.10	0
**PDGF**	1.60	0
**VEGF**	2.30	100

Marked in gray are the molecules with null or reliable MDC in the study participants. IL: interleukin; IL-1RA: interleukin-1 receptor antagonist; bFGF: basic fibroblast growth factor; Eotaxin: CC chemokine subfamily of eosinophil chemotactic proteins; G-CSF: granulocyte colony stimulating factor; GM-CSF: granulocyte–macrophage stimulating factor; IP10: interferon gamma-induced protein 10 kDa; IFγ: interferon gamma; MIP-1α/β: macrophage inflammatory protein-1 alpha/beta; MCP-1: monocyte chemoattractant protein-1; RANTES: regulated on activation, normal T cell-expressed and -secreted; TNF-α: tumor necrosis factor alpha; PDGF: platelet-derived growth factor; VEGF: vascular endothelial growth factor.

**Table 5 ijms-26-06461-t005:** Data from the bead-based immunoassay technique used to determine the inflammatory and angiogenic mediators in the aqueous humor of our study population.

Parameters (pg/mL)	SCG Mean ± SD	T2DMG–DR Mean ± SD	T2DMG + DR Mean ± SD	*p*-Value SCG vs. –DR	*p*-Value SCG vs. +DR	*p*-Value –DR vs. +DR
IL-1β	1.64 ± 0.23	4.7 ± 3.25	7 ± 2.09	0.0142	0.1630	**0.0014**
IL-1Ra	4.03 ± 1.9	5.91 ± 3.67	18.54 ± 0.09	0.0019	0.0002	0.4317
IL-2	1.05 ± 0.49	0.57 ± 0.25	0.41 ± 0.12	0.1572	0.0573	0.3669
IL-6	5.24 ± 1.20	8.88 ± 2.8	11.81 ± 3.32	0.0237	0.0007	0.0575
IL-7	3.53 ± 1.82	3.32 ± 2.03	2.89 ± 1.17	0.8778	0.6605	0.7676
IL-8	0.79 ± 0.08	1.62 ± 0.73	2.85 ± 0.95	0.0016	0.0002	**0.0050**
IL-10	2.87 ± 1.58	3.60 ± 0.94	5.30 ± 2.82	0.5703	0.0784	0.2417
IL-12	12.58 ± 4.92	11.30 ± 4.5	10.81 ± 3.03	0.3709	0.2242	0.7273
IL-13	2.44 ± 1.08	2.24 ± 1.86	2.12 ± 1.85	0.8819	0.8123	0.9289
IL-15	5.15 ± 2.08	8.05 ± 3.12	10.69 ± 2.33	0.0821	0.0023	0.0597
IL-17	4.53 ± 2.11	6.95 ± 2.5	11.14 ± 4.31	0.0119	0.0006	0.1069
bFGF	12.75 ± 5.47	17.3 ± 8.09	26.12 ± 10.18	0.0076	1.94 × 10^−6^	**7.30 × 10^−5^**
GM-CSF	50.58 ± 9.5	74.48 ± 8.99	91.32 ± 11.16	7.91 × 10^−9^	4.21 × 10^−11^	**2.27 × 10^−7^**
IP-10	51.68 ± 28.6	137.33 ± 106.5	205.31 ± 92.18	2.59 × 10^−14^	4.68 × 10^−17^	**1.60 × 10^−13^**
MIP-1β	7.82 ± 2.06	13.89 ± 3.74	18.32 ± 5.15	0.0012	1.67 × 10^−5^	**0.0088**
MCP-1	198.61 ± 59.49	267.53 ± 60.12	310.41 ± 71.28	2.27 × 10^−13^	1.82 × 10^−15^	**2.53 × 10^−11^**
RANTES	2.03 ± 1.16	6.68 ± 4.01	5.56 ± 7.71	0.0003	0.0221	0.1868
VEGF	65.26 ± 20.8	97.36 ± 22.24	118.4 ± 30.16	4.42 × 10^−10^	3.01 × 10^−12^	**2.71 × 10^−8^**

SCG: surrogate control group; T2DM: type 2 diabetes; +DR: with non-proliferative diabetic retinopathy; -DR: without non-proliferative diabetic retinopathy; IL: interleukin; R: receptor; IL-1Ra: interleukin 1 receptor antagonist; bFGF: basic fibroblast growth factor; GM-CSF: granulocyte macrophage colony stimulating factor; IP-10: interferon gamma-induced protein 10; MIP-1b: macrophage inflammatory protein 1 beta; MCP-1: monocyte chemoattractant protein-1; RANTES: regulated on activation, normal T cell-expressed and -secreted protein; VEGF: vascular endothelial growth factor. Molecules showing statistical significance between the corresponding comparative groups are highlighted in bold.

**Table 6 ijms-26-06461-t006:** Inclusion and exclusion criteria for the study population.

Inclusion	Exclusion
Individuals of both sexes aged 40–80 years	Individuals of both sexes, younger than 40 years or older than 80 years
Accurate NPDR diagnosis for the corresponding group of T2DM participants	Other DM or DR type
Non-diabetic healthy individuals for the surrogate comparative group of participants (SCG)	Other ocular diseases and/or comorbidities. Patients receiving local or systemic treatment that may interfere with the study. Eye/laser surgery in the previous 12 months
Precise and complete data included in the clinical history	Clinical history including any diagnosis that did not fit with the study’s purpose
Adequate psycho-physical status for participating in the study	Unfeasibility for having a thorough and complete clinical history. Unable to participate

NPDR: non-proliferative diabetic retinopathy; T2DM: type 2 diabetes mellitus; SCG: surrogate comparative control group; DM: diabetes mellitus; DR: diabetic retinopathy.

## Data Availability

The data presented in this study are available on request from the corresponding author. The data are not publicly available due to institutional privacy regulations.

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
