# Peer review of "Exploring Molecular Signatures Associated with Inflammation and Angiogenesis in the Aqueous Humor of Patients with Non-Proliferative Diabetic Retinopathy"

_ijms, 2025, doi:10.3390/ijms26136461_

Round 1
Reviewer 1 Report
Comments and Suggestions for Authors
This study analyzed the inflammation (INF) and angiogenesis (ANG)-related molecular markers in the aqueous humor (AH) of patients with nonproliferative diabetic retinopathy (NPDR) and revealed the specific molecular characteristics of the early stage of NPDR. The experimental design and writing are good. The article provides important evidence for the early diagnosis and targeted treatment of NPDR, which is in line with the scope of the journal. It is recommended to be accepted for publication after the following details are supplemented.
The specific content is as follows:
- SCG is a non-diabetic patient undergoing cataract surgery. Additional information is needed: Does cataract itself affect AH cytokine levels? Is there evidence that the inflammatory background is no different from that of healthy eyes? It is recommended that authors discuss or cite references for support.
- Mean age of participants was 62 ± 6 years. The ages mentioned by the author do not match those in Table 1. The IOP/CSFT data of SCG and NPDRG were not presented in subgroups according to "-DR" and "+NPDR", which was inconsistent with the statistical description. The grouping logic needs to be unified and subgroup comparisons need to be supplemented. It is recommended to modify Table 2.
- TNF-α, IFN-γ and other 9 molecules were not detected. It is necessary to discuss whether it is due to technical sensitivity (such as MDC is too high) or biological reasons (such as low background level in AH).
- The sample size was estimated using the R software epicalc package, but the specific parameters (effect size, α/β values) were not described. The calculation details need to be supplemented (Method 4.1).
- It is recommended that the authors describe the limitations of this study and prospects for future research in the discussion section.
- Some references are too old and should be deleted.
Reviewer 2 Report
Comments and Suggestions for Authors
Thank you for giving me the opportunity to review your manuscript entitled “Exploring Molecular Signatures Associated with Inflammation and Angiogenesis in the Aqueous Humor of Patients with Non-Proliferative Diabetic Retinopathy.” I very much enjoyed reading your manuscript.
Summary:
This manuscript investigates inflammatory and angiogenic molecular signatures in the aqueous humor (AH) of patients with non-proliferative diabetic retinopathy (NPDR). Through a multicenter, case-control design, the authors analyzed AH samples from 116 cataract surgery patients, categorized into type 2 diabetes mellitus (T2DM) with or without NPDR and non-diabetic controls. Using a multiplex bead-based immunoassay, they quantified 27 cytokines, with 18 reliably detected. The study identified elevated levels of pro-inflammatory and pro-angiogenic mediators—including IL-1β, IL-6, IL-8, IL-15, IL-17, MCP-1, MIP-1β, GM-CSF, VEGF, and IP-10—in NPDR patients, suggesting a distinct intraocular signature. The findings support the potential of AH biomarkers for early DR diagnosis and risk stratification, with implications for personalized treatment strategies.
Comments:
1) Literature review:
The manuscript references a comprehensive array of recent and relevant studies to establish the pathophysiological background of DR. However, the review could benefit from a more concise synthesis of existing cytokine biomarker work in AH, particularly differentiating what this study adds beyond prior vitreous or plasma-based studies.
2) Methods:
The methodology is robust, with clearly defined inclusion/exclusion criteria, adequate sample size justification, and rigorous cytokine quantification using Luminex. However, you may discuss the standardization of OCT parameters and how you addressed the potential inter-platform variability (Heidelberg vs. Zeiss).
